# Thermophysical Property Measurements with the Finite Bar

Christoph Ellenrieder [1,*], André Kaufmann [1], Benedikt Reick [1] and Marcus Geimer [2]

1 Institute for Electromobility, University of Applied Sciences Ravensburg-Weingarten, 88250 Weingarten, Germany
2 Institute of Vehicle System Technology, Karlsruhe Institute of Technology, 76131 Karlsruhe, Germany
* Correspondence: christoph.ellenrieder@rwu.de

**Abstract:** Knowledge of thermophysical properties of materials is important in the design process to meet the ambitious targets with respect to reliability and performance of many modern machinery. In this paper a simple method for the measurements of thermophysical material properties is presented. A bar of the sample material is heated at one end by a constant heat source and temperature sensors on or in the sample material at different locations record the temperature response. In the limit of small Fourier-Numbers the temperature will not rise at the adiabatic end and the comparison to the theoretical curve allows to extract thermophysical data. In the case of large Fourier-Numbers a quasi steady temperature profile in the bar allows to extract all relevant thermophysical properties simultaneously. Apart from the theory some measurement results are presented and the errors due to diabatic boundary conditions are discussed.

**Keywords:** thermophysical properties; heat capacity; conductivity; conduction

## 1. Introduction

The efficient design of cooling structures with modern manufacturing technologies such as 3D metal printing requires the knowledge of the thermophysical properties of the printed material [1]. Different methods for the determination of thermophysical material properties of solid materials are available. Most of the methods are based on temperature measurement and comparison to analytical solutions [2] of the temperature. The methods differ in the sample geometry and the form of applying energy to the sample. The methods can be classified accordingly into an energy pulse as in the laser flash method, fixed boundary temperatures or fixed heat flux at boundaries.

In the laser flash method [3], a sample of the material is exposed to a laser flash on one side and the temperature rise of the probe is used to determine the thermal diffusivity, see Gobrecht et al., 1970 [4] and the corresponding standards and references (ASTM E2585-09 [5], ASTM E1461-13 [6]).

In the Searls bar method [7], a linear sample of the material is kept at fixed temperature at both ends. The temperature and the applied heat are measured. This allows to determine the thermal conductivity.

In the transient line source method [8], a probe of small diameter is inserted into a cylindrical sample of much larger diameter. The applied heat and the temperature are monitored over time. The logarithmic temperature rise allows determining the thermal conductivity.

In this paper, a rather simple method for the determination of thermophysical properties based on one dimensional conduction is presented. It differs from the previously mentioned methods by a cost-effective realization without any potentially dangerous equipment like a laser and no necessity of a regulated heat sink. The boundary conditions are defined by a constant heat flux at one side of the bar and a diabatic condition at the other end of the bar. Monitoring the temperature at different locations and comparison to the analytical solution for small and large Fourier numbers allows the determination of the

thermophysical properties. In the literature, analytical solutions for the one dimensional bar are available for several boundary conditions, see Carslaw [2,9]. The time dependent analytical spacial temperature profile is then typically given by an infinite series.

An alternative and mathematically simple way is to restrict oneself to the asymptotic solutions of very small and very large physical time scales. In the limit of small Fourier-Numbers at small physical time scales when the heat flux has not caused a temperature rise at the adiabatic end, the analytical solution of the semi-infinite body can be applied. The temperature evolution can then be used to determine the product of thermal conductivity, density and heat capacity. In the case of large Fourier-Numbers, a quasi-steady spacial temperature profile develops. It is only shifted by an offset temperature. The change of the offset temperature is then used to determine the heat capacity of the sample material and the spacial shape of the temperature curve allows determining the conductivity.

The theoretical considerations are only directly applicable to the measurement when adiabatic boundary conditions are realized. As the reality is diabatic the heat flux to the surrounding can be partially compensated by measuring the temperature while the sample is cooling under the same boundary conditions as it was heated.

Compared to other methods, this method has the advantage of allowing to determine several thermophysical properties, i.e., heat capacity $c_p$ and conductivity $\lambda$ in one measurement for a wide range of conductivities. In the case of low conductivities, the small Fourier-number limit, the thermal diffusivity $a$ can be determined and in the cases of large conductivities, the large Fourier-number limit, heat capacity $c_p$ and conductivity $\lambda$ can be determined. Measurement results are compared to available data sheet values [10–12] and have not been compared with other measurements using the same sample material.

## 2. Physical Foundations

Detailed derivations of the heat conduction in the one dimensional bar can be found in standard textbooks, see Incropera et al., 2007 [13]. In this section, only the relevant connections to the measurement are stated. Fourier's law states that the specific heat flux $\dot{q}$ is proportional to the temperature gradient in the material $\nabla T$ and the conductivity of the material $\lambda$.

$$\vec{\dot{q}} = -\lambda \nabla T(x,t). \tag{1}$$

The temporal change of temperature in a body is described by the law of temperature diffusion, with the diffusion coefficient $a = \lambda/(c_p \cdot \rho)$ and the second derivative in space $\nabla^2 T$.

$$\frac{\partial}{\partial t} T(x,t) = a \nabla^2 T(x,t) \tag{2}$$

Assuming constant diffusivity $a$ and constant conductivity $\lambda$ in the process, Fourier's law (Equation (1)) and the temperature diffusion equation (Equation (2)) can be combined to find an equation for the diffusion of the specific heat flux.

$$\frac{\partial}{\partial t} \vec{\dot{q}}(x,t) = a \nabla^2 \vec{\dot{q}}(x,t) \tag{3}$$

Stating the equation in terms of heat flux, simplifies the application of the experimental boundary conditions when solving the equation.

### 2.1. Heat Conduction in the One Dimensional Bar

When the width and height of the bar are significantly smaller than the length of the bar, the mathematical problem becomes one dimensional with respect to the longitudinal direction of the bar. A sketch of the geometry and the expected temperature profile is given in Figure 1. With the diabatic BC, no heat is transferred to the ambient. The absolute temperature of the bar will continuously, for this reason, only the temperature differences $\Delta T$ along the bar are shown.

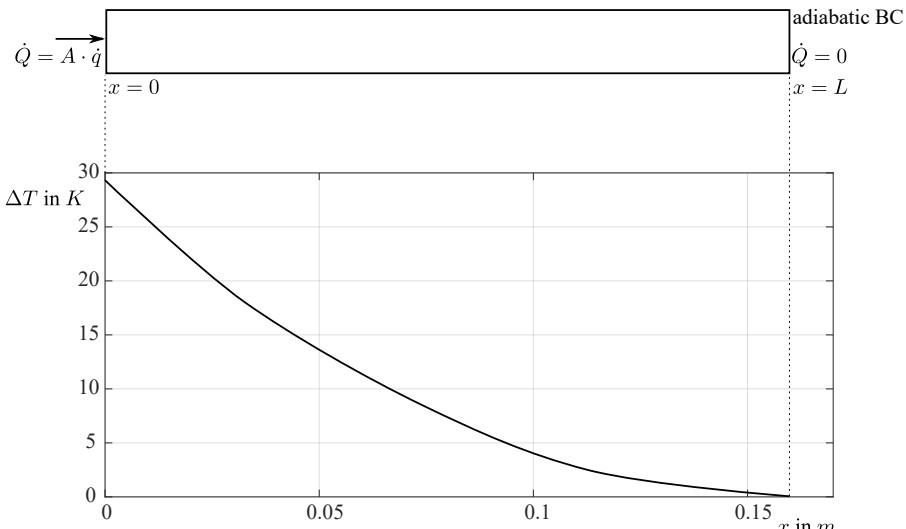

**Figure 1.** Sketch of the bar with the boundary conditions and the expected temperature profile.

The situation can be further simplified when the bar is considered to be semi-infinite, i.e., that the bar is of infinite length. Then the boundary conditions need to be specified at $x = 0$ and $x \to \infty$. Equation (3) can be made non-dimensional. The arising factor and the boundary conditions

- $\dot{q}(x,t)|_{x\to\infty} = 0$
- $\dot{q}(x = 0, t > 0) = \dot{q}_0 = const.$
- $\dot{q}(x, t = 0) = 0$

suggest a variable substitution $\eta = x/(2\sqrt{at})$ in the same way as the standard substitution in the semi-infinite body (see Incropera [13]). This yields an analytical solution for the specific heat flux $\dot{q}$. Integration of the specific heat flux yields the temperature profile [14] in the semi-infinite bar.

$$T(x,t) = T_0 + \frac{2\dot{q}_0\sqrt{at}}{\lambda} \int_{\frac{x}{2\sqrt{at}}}^{\infty} erfc(\mu)d\mu \tag{4}$$

Here $erfc(\mu)$ is the error function complement. The temperature rise at the heated end of the bar $\Delta T(x = 0, t)$ follows a square root dependence on time ($\propto \sqrt{t}$). The next step is to consider a bar of finite length. This is possible when the time since the beginning of heating is so small, that the heat flux has not caused the temperature to rise at the adiabatic end of the bar. This condition can be described with small Fourier numbers.

### 2.2. Solution in the Small Fourier-Number Limit

At small physical times, when the Fourier-Number $Fo = at/L^2$ is significantly smaller than 1, the temperature rise at the adiabatic end is almost zero. This requires the integral $\int_{\frac{L}{2\sqrt{at}}}^{\infty} erfc(\mu)d\mu$ in Equation (4) to vanish. The measured temperature profile can then be compared to the analytical temperature profile to determine the factor $2\dot{q}_0\sqrt{a}/\lambda$.

The temperature rise at the heated end is then given by

$$\Delta T_{sF}(x = 0, t) = \frac{2\dot{q}_0\sqrt{t}}{\sqrt{\lambda\rho c_p}\sqrt{\pi}} \tag{5}$$

since the integral $\int_0^\infty erfc(\mu)d\mu = 1/\sqrt{\pi}$ is constant. With a specific heat flux $\dot{q}_0$ at the heated end ($x = 0$), it is sufficient to measure the temperature at the heated end to de-

termine the factor $\lambda \rho c_p$. When the specific heat capacity $c_p$ and density $\rho$ are known, the conductivity can be extracted from the factor.

$$\lambda = \frac{1}{\rho c_p} \frac{4 \dot{q}_0^2}{\pi} \frac{t}{\Delta T_{sf}^2(x = 0, t)} \tag{6}$$

The equation for the temperature distribution (Equation (4)) can be simplified when the temperature between the heated and the adiabatic end of the bar $\Delta T_{sF}$ is known.

$$T(x, t) = T_0 + \Delta T_{sF} \sqrt{\pi} \int_{\frac{x}{2\sqrt{at}}}^{\infty} erfc(\mu)d\mu \tag{7}$$

This equation is more suited for the post-processing of the experimental data, since $\Delta T_{sF}$ can be obtained directly by measurements. When temperature is monitored at several positions along the bar, it is possible to fit the curve to the temperature Equation (7) with the product by variation of $\eta = x/(2\sqrt{at})$ with known time $t$ and position $x$ to extract the thermal diffusivity $a$.

### 2.3. General Solution of Time-Dependent Conduction for the Finite Bar

The general temperature distribution in a bar of length $L$ is given by Carslaw and Jaeger 1959 [9] by

$$T(x, t) = \frac{1}{L} \int_0^L T(0, t)dx + \frac{2}{L} \sum_{n=1}^{\infty} \exp \frac{-n^2 \pi^2 at}{L^2} \cos \frac{n\pi x}{L} \int_0^L T(x, 0) \cos \frac{n\pi x}{L} dx \tag{8}$$

and is the base for developing thermophysical properties measurement as the flash method [15]. This analytical solution is remarkable with respect to the splitting of the solution in time and space. The solution can be further simplified for large physical time scales.

### 2.4. Solution in the Large Fourier-Number Limit

At large physical time scales, when the Fourier-Number $Fo = at/L^2$ is many times greater than 1, the temperature profile becomes quasi-steady with respect to space.

The analytical solution can be derived, considering the heat source term $\dot{Q}$ to be a Dirac function in space on the left side of the bar ($x = 0$). This can also be viewed as a boundary condition on the left side, with $\frac{\partial T}{\partial x} = -\frac{\dot{Q}}{\lambda A}$ with $\lambda$ being the unknown thermal conductivity and $A$ the surface on the left side of the bar. The shape of the temperature profile from left to right remains similar, it is simply shifted by the increase in the average bar temperature. As the temperature difference $\Delta T_{lF} = T(x = 0, t) - T(x = L, t)$ from the left side of the bar $T(x = 0, t)$ to the right side of the bar $T(x = L, T)$ remains constant, it decouples from the temporal change in temperature.

$$T_{as}(x, t) = \frac{\dot{Q} \cdot t}{mc_p} + \Delta T_{lF} f(x) \tag{9}$$

We refer to this temperature profile as the asymptotic temperature solution $T_{as}(x, t)$ given by Equation (9). The time-dependent component is given by the constant power heat source $\dot{Q}$. The spatial distribution of the temperature is time independent and represented by $f(x)$. This leads to simple derivatives with respect to time and space. The temporal derivative is simply the heating power divided by mass and heat capacity. The second spatial derivative consists of the temperature difference $\Delta T_{lF}$ times the second spatial derivative $\frac{\partial^2 f(x)}{\partial x^2}$.

$$\frac{\partial T(x, t)}{\partial t} = \frac{\dot{Q}}{mc_p} \qquad \frac{\partial^2 T}{\partial x^2} = \Delta T_{lF} \frac{\partial^2 f(x)}{\partial x^2} \tag{10}$$

Placing the terms into the temperature diffusion Equation (2) yields the following relation.

$$\frac{\dot{Q}}{m c_p} = a \Delta T_{lF} \frac{\partial^2 f(x)}{\partial x^2} \tag{11}$$

With a constant heat source $\dot{Q}$, $f(x)$ is a simple second order polynomial. For this reason, the asymptotic temperature distribution $T_{as}$ must follow the following equation with unknown constant $c_1$ and $c_2$.

$$T_{as}(x,t) = T(0,t) + c_1 \cdot \left(\frac{x}{L}\right) + c_2 \cdot \left(\frac{x}{L}\right)^2 \tag{12}$$

The unknown constants can be obtained with the boundary conditions on the left and the right end of the bar. Next to the temperatures at both ends of the bar, the lack of heat flux at the right end of the bar gives an additional condition for the spatial derivative of the temperature profile.

$$\frac{\partial T(x,t)}{\partial x}\Big|_{x=L} = 0 \tag{13}$$

Applying the boundary condition leads to a simple second order polynomial for the temperature profile in the bar.

$$T_{as}(x,t) = T(0,t) + \Delta T_{lF} \cdot \left(\frac{x}{L}\right)\left(\left(\frac{x}{L}\right) - 2\right) \tag{14}$$

Temperature measurements at both ends of the bar are sufficient to determine $\Delta T_{lF}$ and to verify that the asymptotic state is reached. Then the temporal change in temperature can be used to determine the specific heat capacity $c_p$.

$$c_p = \frac{\dot{Q}}{m \frac{\partial}{\partial t} T(x=0,t)} \tag{15}$$

The temperature difference $\Delta T_{lF} = T(x=0,t) - T(x=L,t)$ at both ends of the bar allows determining the thermal diffusivity $a$ in the large Fourier-Number limit and thereby the conductivity $\lambda$.

$$\begin{aligned} a &= \frac{\dot{Q}}{m c_p} \frac{L^2}{2 \Delta T_{lF}} \\ \lambda &= a \rho c_p \end{aligned} \tag{16}$$

The obtained values can be compared to the values determined in small Fourier-Number limit.

### 3. Experimental Setup

The previously described theory can be applied to an experimental setup which consists of a metal bar with length to width ratio ($l/w$) larger than ten. This allows the temperature distribution to be almost one-dimensional when the heat source is heating the entire height on one side of the bar. The bar shown in Figure 2 is heated on the left side ($x = 0$) with a constant power source homogeneously over the entire height. In this setup, an electric resistance heater cartridge is used as a heat source. The geometry of the sample was chosen to satisfy the operating conditions of the heating cartridge. This includes a hollow cylinder to house the heating cartridge. The minimal allowed wall thickness surrounding the heating cartridge defines the height of the bar. The heating cartridge is powered by an external constant power source. Monitoring of current and voltage allows controlling the power in a window with an error lower than 1%.

The sample used here consists of a bar with a total length $l$ of 167 mm, a width $w$ of 40 mm and a thickness of 8 mm. The thickness of 8 mm was chosen to match the diameter of the heating cartridge ($d = 8$ mm). As the heating cartridge requires an absorbing wall thickness of 4 mm, the heated end of the bar was shaped in the form of a cylinder of 16 mm in diameter.

The bar was instrumented with PT100 (Class A) temperature sensors with 2.3 mm in height in 2 mm in width. The temperature sensors have a response time of 0.2 s. Seven temperature sensors were placed on the bar at a distance of 25 mm. The data acquisition system was a self developed PCB using MAX31865 RTD-to-Digital converters.

The heating cartridge power was current controlled using an Aim-TTi CPX400 laboratory power supply.

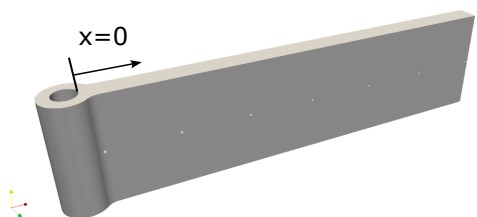

**Figure 2.** One side heated one dimensional bar with temperature measurement.

Temperature recording with a frequency of 1 Hz is typically sufficient to capture all transient effects. The electric heating is triggered a short time after the recording is started. Since the temperature should be uniform at the beginning of the measurement, this allows to verify the calibration of the temperature sensors. This results in no offset error and only temperature dependent. One sample (AL7075) is measured to show the proof of concept.

### 3.1. Characteristic Time Windows of the Measurements

Before the measurement is performed, uniform temperature of the bar must be ensured. This is done by keeping the sample at room temperature for several hours before the measurement is performed. Furthermore, this offers the advantage of keeping the error due to diabatic effects small in the small Fourier-Number limit. The measurements itself can be divided into three phases:

1. Sensor alignment and calibration
2. Constant heating phase
3. Cooling phase

The necessary measurements for the determination of the thermophysical properties take place during the constant power heating phase. Measurement without heating before and afterward are necessary to calibrate the sensors and correct the diabatic effects. The heating phase itself can be divided into three characteristic phases:

1. The phase of small Fourier-Numbers, where the temperature rise at the adiabatic end has not started (Small Fo window).
2. Transient phase in which an asymptotic temperature profile is slowly established.
3. The phase of large Fourier-Numbers, where the asymptotic temperature profile is stable (Large Fo window).

Figure 3 shows the temperature measurements without calibration corrections on the sample bar AL7075. On the left side, the small time window for the small Fourier-Number limit is marked. In the middle, the time window for the large Fourier-Number limit and the different curves for the sensors can be seen. The processed data of the cooling phase lies in the cooling window, which ranges approx. from 1000 s up to the end of the measurement.

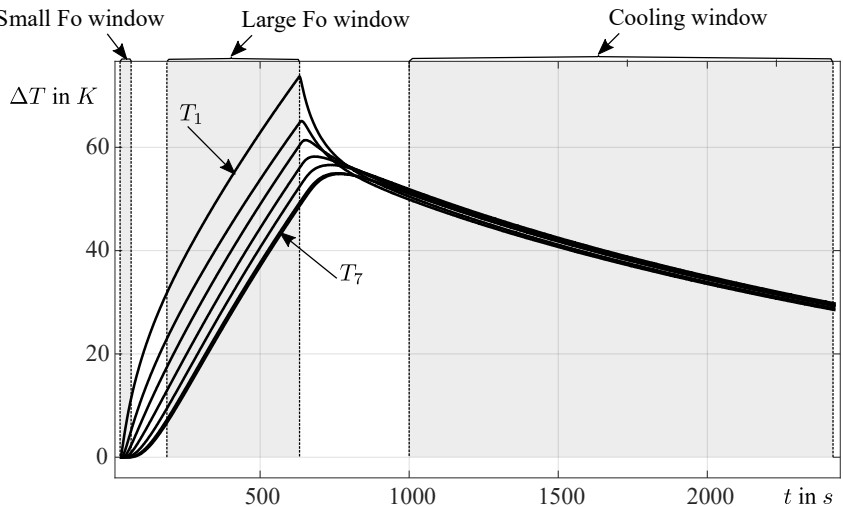

**Figure 3.** Temperatures at the different locations of the bar, $T_1$ is next to the heating cartridge, $T_7$ at the adiabatic end of the bar.

### 3.2. Temperature Dependence of Thermophysical Properties

Specific heat capacity $c_p$ and conductivity $\lambda$ changes with temperature [16,17]. The changes in a limited heating range of less than 50 K at temperatures between 0 °C and 150 °C tend to be lower than the uncertainty of most methods. The presented method can be extended to a procedure, where the changes in thermal diffusivity due to temperature change can be extracted by locally fitting the heating slopes to limited temperature ranges in the large Fourier-Number limit. Application of the small or the large Fourier-Number limit can be adapted by the heating power and the length of the bar. This extension is subject of current research.

### 3.3. Errors of the Measurement Procedure

The errors in the measurement procedure are of multiple origin. Major errors are expected to arise from

- geometric difference to the ideal mathematical one dimensional geometry
- inclusion of the heating cartridge of a different, unknown material
- diabatic boundary conditions where adiabatic boundary conditions are assumed
- geometric extension and geometric position of the temperature sensors
- temperature measurement error due to calibration and measurement equipment
- precision of the power supply and heating loss to the connecting wires

The errors can be partially compensated.

### 3.4. Correction of Systematic Errors

For the geometric difference, the bar is separated into the heating cartridge (H), the hollow cylinder (C) and the bar (B). In the small Fourier-Number limit, it must be ensured that the characteristic time scales $\tau$ of the heating cartridge with diameter $d_H$ and the hollow cylinder are small compared to the characteristic timescale of the bar.

$$
\begin{aligned}
\tau_H &= \frac{d^2}{16 a_H} \ll & \tau_B &= \frac{L^2}{a_S} \\
\tau_C &= \frac{(d_a - d_i)^2}{4 a_S} \ll & \tau_B &= \frac{L^2}{a_S}
\end{aligned}
\tag{17}
$$

This requires the knowledge of the thermal diffusivity ($a$) of the sample ($S$) material and the cartridge for the first condition and can only be established a posteriori. The second condition is easily verified when the wall thickness of the hollow cylinder is significantly smaller than the length of the bar. The heating of the cartridge $\dot{H}_H$ and the heating of the

hollow cylinder $\dot{H}_C$ can then be subtracted from the electric heating power $P_{el}$. The diabatic contribution should remain small as the temperature difference to the ambient is small during this phase.

The Enthalpy change of the heating cartridge and the hollow sample cylinder at the end can be simply estimated by the temperature change.

$$\dot{H}_H = m_H c_{p,H} \frac{\partial T_H}{\partial t} \qquad \dot{H}_C = m_C c_{p,C} \frac{\partial T_C}{\partial t} \tag{18}$$

In the large Fourier-Number limit, when the asymptotic temperature profiles are established, the enthalpy balance of the experimental setup can be established.

$$\dot{H}_H + \dot{H}_C + \dot{H}_B + \dot{Q}_L + P_{el} = 0 \tag{19}$$

The heat loss due to the diabatic condition $\dot{Q}_L$ can be modeled by assuming a convective heat transfer to the ambient temperature $T_A$.

$$\dot{Q}_L = \alpha A (T_A - T_B) \tag{20}$$

The necessary coefficient for the convective heat transfer $\alpha A$ can be extracted from the cooling phase with $P_{el} = 0\,\text{W}$. When the energy is equally distributed within the bar, all temperature sensors show the same temperature. Then the bar can be considered as a thermal block capacity and the analytic expression for the temperature decrease $T_{cool}(t)$ can be fitted to the data to estimate the time constant $\tau$.

$$T_{cool}(t) = T_a + (T_0 - T_a) e^{-(t - t_{0c})/\tau} \tag{21}$$

Here $T_a$ is the ambient temperature surrounding the block and $T_0$ is the temperature at the starting time $t_{0c}$ of the block capacity cooling process. Using the estimated heat capacity $c_p$ and the relation for the characteristic time $\tau = (m c_p)/(\alpha A)$, the heat flux to the surrounding of the bar can be estimated by

$$\dot{Q} = \frac{m c_p}{\tau} (T_{cool}(t) - T_a) \tag{22}$$

Measurements with insulating materials such as ceramic fiber insulation (thermal conductivity $\lambda$ of $0.07\,\text{W/mK}$) and different heating power revealed an unwanted effect of transient heating of the insulation material with an undefined heat loss compared to free convection. Therefore, it is easier to compensate for the free convection heat loss than compensating for the retained heat in the non-ideal insulation.

## 4. Results

The experimental results are discussed for the different phases of the experiment. In this study, the bar was heated with a power of $P_{el} = 20\,\text{W}$.

### 4.1. Results in the Small Fourier-Number Limit

In the small Fourier-Number limit, the temperature increases at first near the heating cartridge and then diffuses to the adiabatic end of the bar. The temperature measurements for the small Fourier-Number can be extracted by the corresponding criteria: The beginning is defined by the temperature rise in the sensor at the heated end of the bar ($x = 0\,\text{W}$) and the end of the phase is defined by a starting temperature rise in the diabatic end of the bar ($x = L$).

For selected times of small Fourier-Numbers ($Fo$ = 0.022, 0.035, 0.046, 0.059, 0.071, 0.083, 0.096) the temperature measurements at the sensor locations are given in Figure 4. The temperature measurements can be fitted to Equation (7) to determine a value for the thermal diffusivity $a$. Fitted curves are shown by dashed lines. For the sample Material the

curve fitting resulted in a value of $a = 3.8 \cdot 10^{-5}$ m$^2$/s for the thermal diffusivity. This is in the range of the values found in different data-sheets (Table 1).

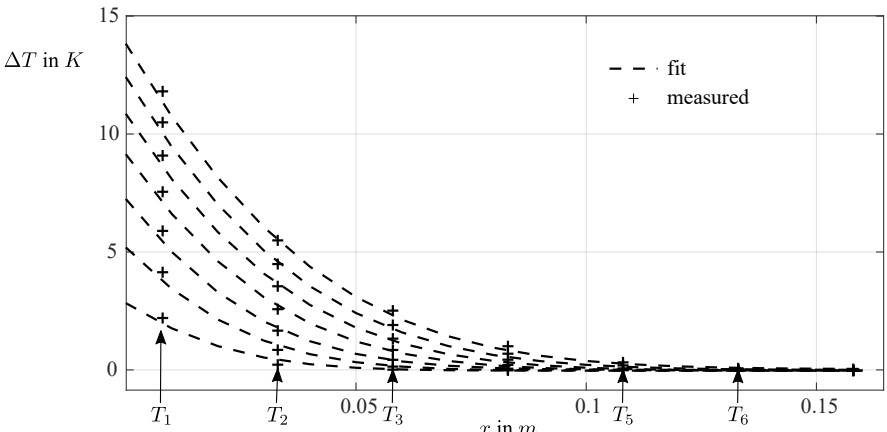

**Figure 4.** Measured temperatures at different times at the sensor locations are given by (+), the fitted curve to the temperature (Equation (4)) is given by the dashed curve (–).

**Table 1.** Results for the thermophysical properties obtained in the large Fourier-Number limit.

| Sensor | $c_p$ in J/kgK | $a \cdot 10^{-5}$ in m$^2$/s | $\lambda$ in W/mK |
|---|---|---|---|
| 1 | 1321.5 | 4.507 | 160.82 |
| 2 | 1319.9 | 4.513 | |
| 3 | 1319.1 | 4.515 | |
| 4 | 1313.7 | 4.534 | |
| 5 | 1317.2 | 4.522 | |
| 6 | 1321.5 | 4.507 | |
| 7 | 1335 | 4.461 | |
| Mean | 1321.1 | 4.51 | 160.82 |
| Datasheet [12] | 862 | 5.38–6.63 | 130–160 |

The temperature $T(x = 0, t)$ rise measured at the origin is shown in Figure 5. By fitting the constants, it can be clearly seen that the measurements correspond to the analytical curve as described by Equation (5). In this case, the fit leads to a value of $\lambda \rho c_p = 7.2 \cdot 10^8$ kg$^2$ m/K$^2$s$^5$. This value differs from the literature value ($3.5 \cdot 10^8$ kg$^2$ m/K$^2$s$^5$) [12] due to the uncertainty of the fit that depends critically on the exact start of the heating time. Knowledge of the specific heat flux $\dot{q}_0$ allows determining the product $\lambda \rho c_p$. Even if the density $\rho$ is known, it is necessary to measure the specific heat capacity $c_p$ before the conductivity $\lambda$ can be computed.

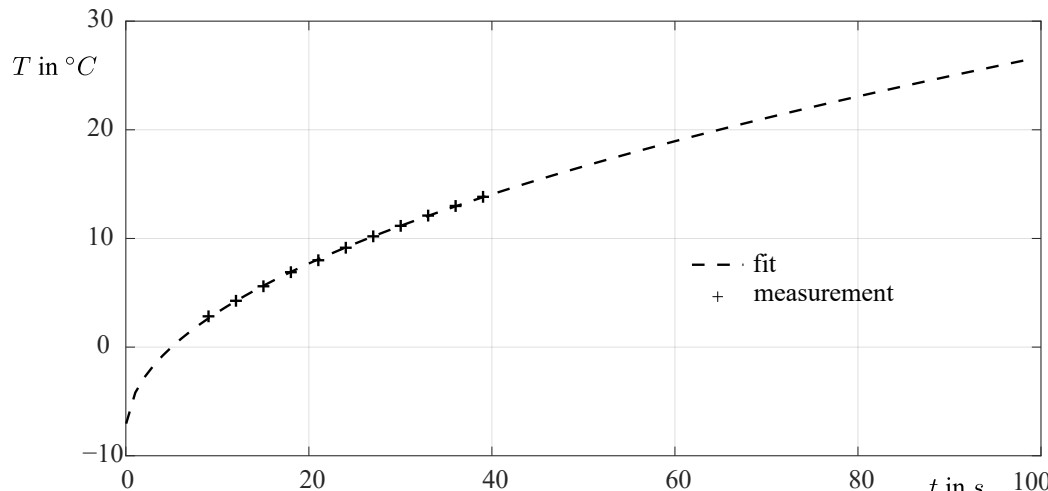

**Figure 5.** Measured temperature of the first sensor as a function of time (+), the fitted curve to the temperature (Equation (5)) is given by the dashed curve (–).

### 4.2. Results in the Large Fourier-Number Limit

The beginning of the large Fourier-Number window can be identified as the time when the first and the last temperature sensor have the same slope $\frac{\partial T_i}{\partial t}$. In the ideal adiabatic case this window would end with the heating phase. In theory, heating curves should be linear with time. Due to diabatic effects at temperatures higher than ambient, heat is lost, and the temperature profiles bend to lower temperature rise. Therefore, the large Fourier-Number window should be limited to times when the heat loss is still significantly smaller than the heat supplied by the heating cartridge.

This can be verified by ensuring that the temperature profile matches the asymptotic solution given by Equation (14). The asymptotic profiles can be superposed by subtracting the offset temperature at the adiabatic end of the bar $(T(L, t))$. The temperature profiles are shown in Figure 6 for selected times. Since the profiles almost superpose, the temperature rise of every sensor has the same temporal change. Figure 7 shows the temperature rise of the different sensors and the corresponding linear fits, starting with the "Large *Fo* window" at $t = 100$ s. By Equation (16) it is directly possible to determine the thermal properties of the material given in Figure 7 when the diabatic corrections are not applied.

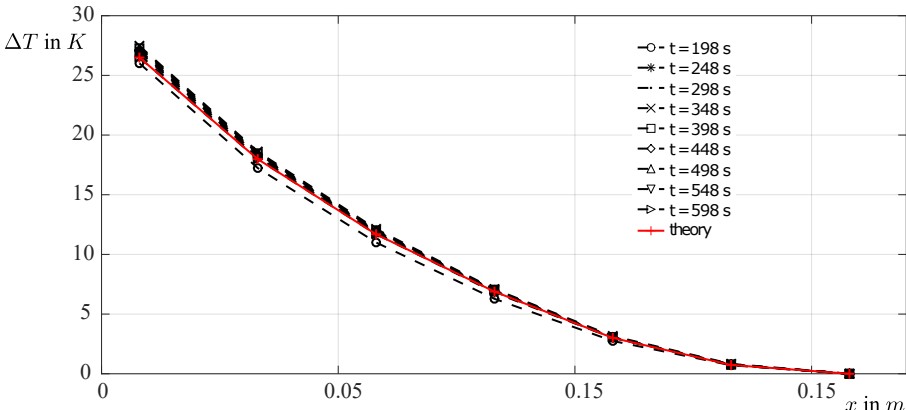

**Figure 6.** Temperature profiles shifted by the offset temperature at different times of the measurement in the large Fourier-Number limit.

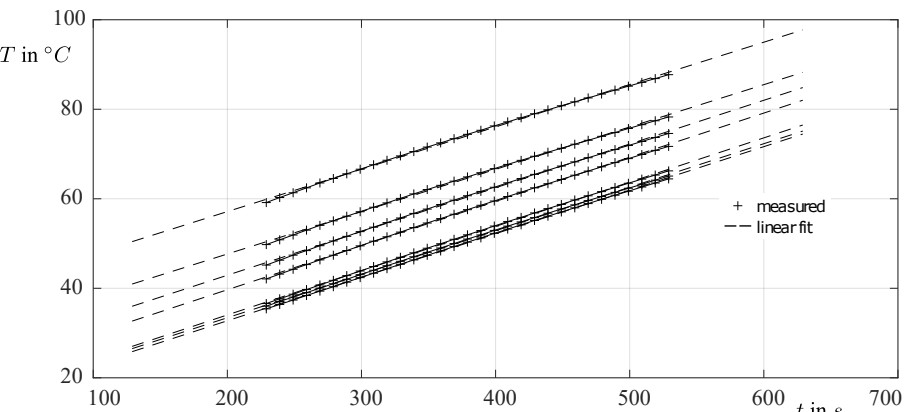

**Figure 7.** Temperatures of the different sensors in the large Fourier-Number window and the corresponding linear fit.

### 4.3. Estimation of Errors Due to Diabatic Effects

The error due to diabatic effects can be estimated by the cooling curve given in Equation (22). In this case, the estimated time constant is $\tau = 2583\,\text{s}$ computed by the fit of the logarithmic temperature gradient of the cooling curve.

The heat loss to the ambient is shown in Figure 8. Compared to the heating power of $P = 20\,\text{W}$ this accounts for an error of roughly 10%. This error transfers to the computation of the heat capacity since the temperature slope would be steeper if the heat losses do not occur. The error can be compensated by correcting the electric power as suggested in Equation (19).

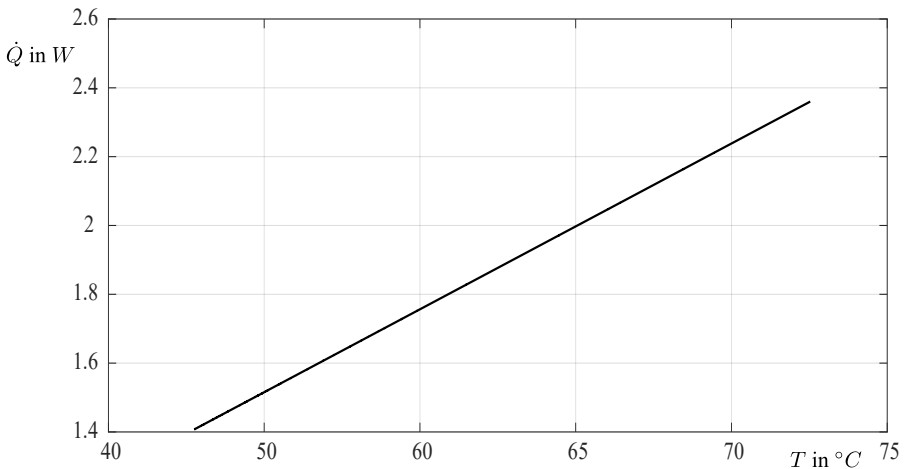

**Figure 8.** Heat loss estimated by the cooling curve.

### 5. Discussion

Results for the determination of material properties with the temperature rise in the small Fourier-Number limit at the origin are prone to error due to uncertainties in the fit of the product. However, fitting the curve with the temperature difference to several sensors shows results comparable to literature values and is more reliable. Since the temperature difference to the ambient temperature is still small, the evaluation of properties in the small Fourier-Number limit leads to smaller errors due to heat transfer to the ambient compared to the results obtained from the large Fourier-Number limit.

Fitting the curves in the large Fourier-Number limits give acceptable values compared to literature values due to the stable shape of the curves. Due to the temperature rise,

the heat transfer to the ambient is larger than in the small Fourier-Number limit and has therefore a larger error due to heat transfer.

The error in the small Fourier-Number limit can be reduced further by increasing the induced heat flux. This leads to a higher temperature difference in the bar at the same timescale and therefore to a smaller error for the determination of the thermal diffusivity $a$.

If the heat flux is increased, the time range with stable temperature profiles in the large Fourier-Number limit is decreased and the error due to heat loss to the ambient increases. Performing the experiment in an environment that allows to control the temperature surrounding the object can decrease the error due to heat flux. An alternative method to decrease the error due to heat flux is to control the ambient temperature to the average temperature of the bar.

## 6. Conclusions

Due to the complexity and cost of THW and TAB thermophysical property measurements, a different cost-effective and simple measurement procedure was developed, using only PT100 sensor elements, a heating cartridge and electronic for data acquisition and power control. As analytical mathematical solutions are readily available for the one dimensional bar, this geometry was selected for the property measurement. The analytical series solution given by Carslaw [9] supplies solutions for all physical times. More simple and physically instructive solutions exist in the small and large Fourier-Number limit. The limit solutions and methodology to extract the material properties are presented in detail. The geometry of the probe was adapted for the heating with a simple electric resistance and the instrumentation with temperature sensors. The measurement procedure consists of heating the bar with the electric resistor. Temperature measurement prior to heating and after heating allows calibration of the sensors. The heating phase can be divided into small Fourier-Number limit, transient phase and large Fourier-Number limit. Separate evaluation of small and large Fourier-Number limits allow determining the thermophysical properties (thermal diffusivity, capacity, and conductivity). Systematic and random errors are briefly discussed. Detailed discussion and the compensation of errors are the subject of a planned subsequent publication.

**Author Contributions:** Data curation, C.E. and A.K.; Formal analysis, A.K.; Methodology, A.K.; Project administration, C.E.; Software, C.E. and A.K.; Supervision, B.R. and M.G.; Writing—original draft, A.K.; Writing—review & editing, C.E., B.R. and M.G. All authors have read and agreed to the published version of the manuscript.

**Funding:** This Project is supported by the Federal Ministry for Economic Affairs and Climate Action (BMWK) on the basis of a decision by the German Bundestag.

**Institutional Review Board Statement:** Not applicable.

**Informed Consent Statement:** Not applicable.

**Data Availability Statement:** Measurement data is available from the authors upon request.

**Conflicts of Interest:** The authors declare no conflict of interest.

## Abbreviations and Nomenclature

The following abbreviations are used in this manuscript:

| | |
|---|---|
| BC | Boundary condition |
| DUT | Device under test |
| THW | Transient hot wire method |
| THB | Transient hot bridge method |



| Symbol | Meaning | SI-Unit |
|--------|---------|---------|
| $A$ | Cross section of the bar | $m^2$ |
| $c_p$ | Specific heat capacity | - |
| $a$ | thermal diffusivity | $m^2/s$ |
| $Fo$ | Fourier-Number | - |
| $L$ | length of DUT | m |
| $\dot{q}$ | specific heat flux | $W/m^2$ |
| $\dot{Q}$ | specific heat | W |
| $T$ | Temperature | K |
| $x$ | Position along the DUT length | m |
| $\lambda$ | Thermal conductivity | W/mK |
| $\rho$ | Density | $kg/m^3$ |

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
