# Peer review of "Thermophysical Property Measurements with the Finite Bar"

_applsci, doi:10.3390/app131810371_

Round 1

Reviewer 1 Report

1)  In this paper a simple method for the measurements of thermophysical material properties is presented. Has this method been compared with methods widely used in the literature to verify the accuracy of the method?

2)  The real measurement process of material thermophysical properties is non-adiabatic, so what is the actual measurement error? How to reduce it?

Author Response

1) In this paper a simple method for the measurements of thermophysical material properties is presented. Has this method been compared with methods widely used in the literature to verify the accuracy of the method?

Response 1:

  • Additional motivation has been added
  • Hot wire method has been removed
  • at this point of the study no reference measurements were performed, only datasheet values were used. This information has been added to the paper in the introduction (Line 59+60) as well as compared to results from other papers and datasheets [8-10]

2) The real measurement process of material thermophysical properties is non-adiabatic, so what is the actual measurement error? How to reduce it?

Response2:

  • The error is shown in “correction of systematic errors” – further explanation why the probe is NOT insulated was added to this section as well ( Line 273-277)
  • It is easier to compensate for the free convection heat loss analytically (Line 259ff)

Reviewer 2 Report

Introduction

Your methodology for measuring the thermo-physical properties is applicable to solid materials, and this should be highlighted. In the Introduction you briefly mention a few measuring techniques applicable to gases, and then propose your methodology. You should mention other measuring techniques available for solids, and then explain in what respect is the methodology that you propose here novel or original in comparison to other methodologies available for solids; methodologies available for gases can be mentioned for completeness, but should not be used as a comparison for the present methodology, which is applicable to solids and not to gases.

Methodology

You neglect the effect of temperature on the thermal properties of the bar, this assumption should be justified properly. You refer to a semi-infinite bar, how long does an actual bar need to be to approach such an approximation? When can you actually use the model for a semi-infinite bar in an actual experimental setup with a bar of finite length? Thermal expansion effects should be estimated and discussed.

Experiment

Since you are proposing a new measuring methodology, I suggest to expand the description of the experimental setup by providing enough information to allow repeatability: CAD with dimensions, all details regarding the heater used, precise location of the thermocouples (with details of calibration), details of the DAS,…. You provide measurements for one sample as a proof of concept; adding more examples would make the paper stronger.

Why not using thermal insulation during the tests?

Measuring errors should be discussed more extensively, error-bars should be included in all figures reporting measured data, errors should be included in the table reporting the results.

Adding a few more references would increase the visibility of the paper.

Author Response

Point 1:
Introduction

Your methodology for measuring the thermo-physical properties is applicable to solid materials, and this should be highlighted. In the Introduction you briefly mention a few measuring techniques applicable to gases, and then propose your methodology. You should mention other measuring techniques available for solids, and then explain in what respect is the methodology that you propose here novel or original in comparison to other methodologies available for solids; methodologies available for gases can be mentioned for completeness, but should not be used as a comparison for the present methodology, which is applicable to solids and not to gases.

Response 1:

  • Method for gases has been removed
  • More methods have been added to the state of the art

Point 2:
Methodology

You neglect the effect of temperature on the thermal properties of the bar, this assumption should be justified properly. You refer to a semi-infinite bar, how long does an actual bar need to be to approach such an approximation? When can you actually use the model for a semi-infinite bar in an actual experimental setup with a bar of finite length? Thermal expansion effects should be estimated and discussed.

Response2:

  • Added explanation regarding the thermal dependency of the properties to new subsection (Temperature dependence of thermophysical properties, Line 222)
  • Thermal expansion is irrelevant (23,4µ/K for AL7075), this results in a total length variation of the probe of 27µm for a ΔT of 70 K

Point 3: Experiment

Since you are proposing a new measuring methodology, I suggest to expand the description of the experimental setup by providing enough information to allow repeatability: CAD with dimensions, all details regarding the heater used, precise location of the thermocouples (with details of calibration), details of the DAS,…. You provide measurements for one sample as a proof of concept; adding more examples would make the paper stronger.

Why not using thermal insulation during the tests?

Measuring errors should be discussed more extensively, error-bars should be included in all figures reporting measured data, errors should be included in the table reporting the results.

Response3:

  • Geometry information of the bar has been added to the paper (Line 184ff)
  • Position of thermocouples described (Line 190)
  • DAQ System Information added (Line 190-194)
  • The use of insulation brings further problems, as this will heat up the insulation material. The compensation for the diabatic heat loss due to convection yielded better controllable results– Explanation has been added to (Line 259ff) and (Line 273-277)

Point 4:

Adding a few more references would increase the visibility of the paper.

Response 4:

  • Added further introduction with additional references, as well as different methods

Reviewer 3 Report

The authors present a relevant investigation with several demonstrations. I strongly recommend the acceptance of this investigation after addressing the following minor comments:

 1- Add a nomenclature table with SI units.

2- Why prefer this technique of thermal measurement? State other ones as examples.

3- The introduction is short. So, the following references are recommended for citation to show the importance of the thermal conductivity of each phase for enhancing the thermal efficiency of nanofluids:

- A passive control approach for simulating thermally enhanced Jeffery nanofluid flows nearby a sucked impermeable surface subjected to buoyancy and Lorentz forces

- Water thermal enhancement in a porous medium via a suspension of hybrid nanoparticles: MHD mixed convective Falkner's-Skan flow case study

- Numerical treatment of MHD Al2O3-Cu/engine oil-based nanofluid flow in a Darcy-Forchheimer medium: Application of radiative heat and mass transfer laws

- Solutal effects on thermal sensitivity of casson nanofluids with comparative investigations on Newtonian (water) and non-Newtonian (blood) base liquids

- Towards a novel EMHD dissipative stagnation point flow model for radiating copper-based ethylene glycol nanofluids: An unsteady two-dimensional homogeneous second-grade flow case study

- Numerical inspection of two-dimensional MHD mixed bioconvective flows of radiating Maxwell nanofluids nearby a convectively heated vertical surface

- Numerical passive control of alumina nanoparticles in purely aquatic medium featuring EMHD driven non-Darcian nanofluid flow over convective Riga surface

- Importance of exponentially falling variability in heat generation on chemically reactive von kármán nanofluid flows subjected to a radial magnetic field and controlled locally by zero mass flux and convective heating conditions: A differential quadrature analysis

4-Perform a comparison with the existing results (if possible).

5- Improve the physical discussion of the experimental results.

Based on the above comments, I would like to give another chance to the authors to revise the manuscript.

It is good

Author Response

The authors present a relevant investigation with several demonstrations. I strongly recommend the acceptance of this investigation after addressing the following minor comments:

 1- Add a nomenclature table with SI units.

Response 1:

  • Table with symbol, Meaning, SI-Unit was added

2- Why prefer this technique of thermal measurement? State other ones as examples.

Response 2:

  • Added information about State-of-the-Art methods: “flash method” and other impulse methods [3,4], a method with heat source and sink [5], and the “transient line source method” [6] were added
  • Information that the proposed method offers a simple method with easily available lab equipment is emphasized

3- The introduction is short. So, the following references are recommended for citation to show the importance of the thermal conductivity of each phase for enhancing the thermal efficiency of nanofluids:

- A passive control approach for simulating thermally enhanced Jeffery nanofluid flows nearby a sucked impermeable surface subjected to buoyancy and Lorentz forces

- Water thermal enhancement in a porous medium via a suspension of hybrid nanoparticles: MHD mixed convective Falkner's-Skan flow case study

- Numerical treatment of MHD Al2O3-Cu/engine oil-based nanofluid flow in a Darcy-Forchheimer medium: Application of radiative heat and mass transfer laws

- Solutal effects on thermal sensitivity of casson nanofluids with comparative investigations on Newtonian (water) and non-Newtonian (blood) base liquids

- Towards a novel EMHD dissipative stagnation point flow model for radiating copper-based ethylene glycol nanofluids: An unsteady two-dimensional homogeneous second-grade flow case study

- Numerical inspection of two-dimensional MHD mixed bioconvective flows of radiating Maxwell nanofluids nearby a convectively heated vertical surface

- Numerical passive control of alumina nanoparticles in purely aquatic medium featuring EMHD driven non-Darcian nanofluid flow over convective Riga surface

- Importance of exponentially falling variability in heat generation on chemically reactive von kármán nanofluid flows subjected to a radial magnetic field and controlled locally by zero mass flux and convective heating conditions: A differential quadrature analysis

Response 3:

  • Added further details/methods and application examples for this method as well as other methods for determining the thermophysical properties.
  • As the paper proposes a method to characterize solid material probes (high conductive materials) the authors deem the papers proposed - which are mainly about nanofluids even though about heat conduction and cooling – not relevant to this paper

4-Perform a comparison with the existing results (if possible).

Response 4:

  • As this is a proof of concept, the measured data was mainly compared to datasheet values and results from other papers [8-10]

5- Improve the physical discussion of the experimental results.

Response 5:

  • Further details have been added: Temperature dependency (Lines 222ff.)
  • Errors resulting from the current setup and proposal for compensation are given (Lines 232ff, 242ff)

Based on the above comments, I would like to give another chance to the authors to revise the manuscript.

Round 2

Reviewer 2 Report

NA